# Dynamics of *Abies nephrolepis* Seedlings in Relation to Environmental Factors in Seorak Mountain, South Korea

**JiDong Kim** [1] , **Jong-Hwan Lim** [2] **and ChungWeon Yun** [3,*]

1   Baekdudaegan National Arboretum, Bongwa 36209, Korea
2   National Institute of Forest Science, Seoul 02455, Korea
3   Department of Forest Resource, Kongju National University, Kongju 32588, Korea
*   Correspondence: cwyun@kongju.ac.kr; Tel.: +82-10-4312-5745

**Abstract:** We present novel evidence of environmental drivers of seedling density in *Abies nephrolepis*, an alpine and subalpine tree species. Continuous monitoring of natural conditions is required to understand forest ecosystem dynamics. We investigated *Abies nephrolepis* seedling dynamics in relation to biotic and abiotic factors. The survey, which included the measurement of trees and seedlings, was carried out from March to October in 2016 and 2018. Monitoring sites in the coniferous forests of Seorak Mountain were divided into 27 quadrats. We analyzed relationships using simple and multiple linear regression. The majority of *Abies nephrolepis* individuals had a diameter at breast height less than l0 cm, and the number of seedlings increased over the study period. This reflects survival and growth due to successive annual mast seeding events. Aspect direction ($R^2 = 0.201$, $p < 0.05$), rock exposure ($R^2 = 0.364$, $p < 0.001$), canopy openness ($R^2 = 0.322$, $p < 0.05$), herbaceous cover ($R^2 = 0.268$, $p < 0.01$), and basal area ($R^2 = 0.199$, $p < 0.05$) show significant linear relationships with seedling density. Seedling density was positively related to rock exposure, canopy openness, and species richness, and there was a negative relationship between herbaceous cover and basal area ($p < 0.0001$). The relative importance of predictor variables was as follows: Rock exposure (40.3%), canopy openness (30.2%), basal area (13.9%), herbaceous cover (11.5%), and species richness (4.1%). Seedling density was most strongly influenced by the presence of large rocks, which provide shelter from harsh winds and a substrate for moss. We conclude that appropriate canopy openness creates a synergistic relationship. We found a positive association between the *Abies nephrolepis* seedling density in subalpine forests and certain physical environmental factors. Therefore, environmental gradients about the roles of rocks and canopies apply, even in this habitat.

**Keywords:** *Abies nephrolepis*; extreme zone; forest ecology; seedling density

## 1. Introduction

Alpine and subalpine zones, which are considered extreme ecoclines [1], are vulnerable to climate change [2,3]. The rapid retreat of alpine glaciers to higher elevations has been well documented, and there is clear evidence that it is accelerating [4–6]. Despite the importance of regional vulnerability and biodiversity conservation, other forest ecosystems have been relatively poorly studied. Species at high elevations are more sensitive to climate change than those in lowland areas [7–9], because the long lifespan of trees or narrow-range species does not allow for rapid adaptation to environmental changes [10].

To understand long-term forest dynamics, ecological research and data cataloging are necessary [11,12]. Field data are needed for the management of natural forest resources, requiring continuous monitoring

of natural conditions [10,11,13,14]. Understory dynamics, which can be observed via long-term monitoring, provide important ecological information to understand long-term change in forests [15,16].

Long-term monitoring implies a 5- or 10-year interval between study periods; it is thus difficult to examine continuous forest ecosystem dynamic changes during shorter periods [17–21]. Conversely, there are many quantitative studies on the relationships between seed production and climate, topography, and biological factors, but these all have short observation periods, mostly of one year [22]. Many single-site time series studies have proved useful to understand ecosystems. Such single-site time series can be used to observe and understand patterns of natural recruitment.

In particular, in subalpine zones, the regeneration of tree seedlings depends on various environmental, topographical, and climatic factors. The climate at high elevation is characterized by strong seasonal variation with short, moderately warm, and moist summers, and long, extremely cold, and dry winters [23–26]. In general, alpine plants adapted to dry conditions can use water very sparingly [27–29]. Wind conditions have the greatest influence on physiological change and damage of plants at the treeline, and rocks can provide shelter to tree seedlings from wind during the germination and early growth stages [30–33]. The influence of canopy gaps on forest processes such as stand structure, regeneration, and understory dynamics is important [34–36]. Moreover, analysis of the dynamics of seedlings by manipulating neighboring herbaceous cover and microclimate has revealed that protection from bright sunlight, low temperature, and water stress facilitate growth and survival [37–40]. Mast seeding in the understory is a widespread plant reproductive strategy [41] that has been studied in various habitats [42–44].

Seorak Mountain in Korea is isolated from other mountain ranges, and comprises numerous habitats varying in altitude, micro-topography, slopes steepness, and wind strength; the mountain constitutes an extreme environment [45–47]. The survival of alpine plants depends on biodiversity conservation [48,49]. The influence of environmental factors on *Abies nephrolepis* seedlings' density in the Seorak Mountain remains largely unknown. In the present study, *Abies nephrolepis* seedlings and other vegetation in the coniferous forests of Seorak Mountain were monitored for two years. Accordingly, the first objective of this study was to investigate the spatial and temporal patterns of the dynamics of *Abies nephrolepis* seedlings in sub-alpine coniferous forests. Based on the studies cited above, which suggest the hypothesis that seedling density is related to environmental factors; our second objective was to assess whether *Abies nephrolepis* seedlings' density is related to biotic and abiotic environmental factors.

## 2. Materials and Methods

### 2.1. Site Description

Daecheongbong peak (1708 m) in the Seorak Mountain range, near the Baekdudaegan mountain range, is at 38°05′25″ N–38°12′36″ N and 128°18′03″ E–128°26′43″ E; its total area is 398.549 km$^2$. Its vegetation comprises mixed forest or pure forest of evergreen conifers such as *Pinus densiflora*, *Abies holophylla*, *Pinus koraiensis*, and *Abies nephrolepis*. Forest shrubs such as *Lonicera caerulea*, *Crataegus komarovii*, *Taxus caespitosa*, and *P. pumila* are located at the top of the mountain, and alpine plants such as *Anemone narcissiflora*, *Geranium eriostemon*, and *Arctous ruber* are distributed throughout the area. In previous studies of the forests of Seorak Mountain, vegetation [50], population structure [51–56], and vascular plants [57–59] have been investigated. In addition, habitat distribution has been predicted by remote sensing [60]. In South Korea, studies on the seedlings of *Abies koreana* [61–63], *Picea jezoensis* [64,65], and *Abies nephrolepis* [66] have been conducted. Although several studies have been conducted on the ecological characteristics of subalpine conifer species in Japan [36,67,68] and China [69–71], there is a lack of studies in this regard in South Korea.

The soil environmental characteristics of Seorak Mountain are from the National Institute of Forest Science [72]. The soil physical properties are as follows: Coarseness, 35%; composition of 35% sand, 50% silt, and 15% clay; and bulk density, 0.95 g/cm$^3$. The soil chemical properties are as follows:

Acidity, pH 4.4; organic matter, 9.1%; cation exchange capacity, 43.2 cmol$_c$/kg; and exchangeable aluminum, 530 mg/kg. The exchangeable cation contents are Ca$^{2+}$ 0.62 cmol$_c$/kg, Mg$^{2+}$ 0.38 cmol$_c$/kg, K$^+$ 0.2 cmol$_c$/kg, and Na$^+$ 0.18 cmol$_c$/kg.

The climate of Seorak Mountain is shown in the climate diagram (Figure 1), for the eight year period from 2011 to 2018; meteorological data were measured at the Seoraksan 875 weather station (1596 m) [73]: Mean annual temperature, 3.05 °C; mean annual precipitation, 1537.39 mm; highest average monthly precipitation, 398.7 mm in August; mean daily minimum temperature of the coldest month, −23.3 °C; mean daily maximum temperature of the hottest month, 23.7 °C; absolute minimum temperature, −29.7 °C; and absolute maximum temperature, 29.9 °C.

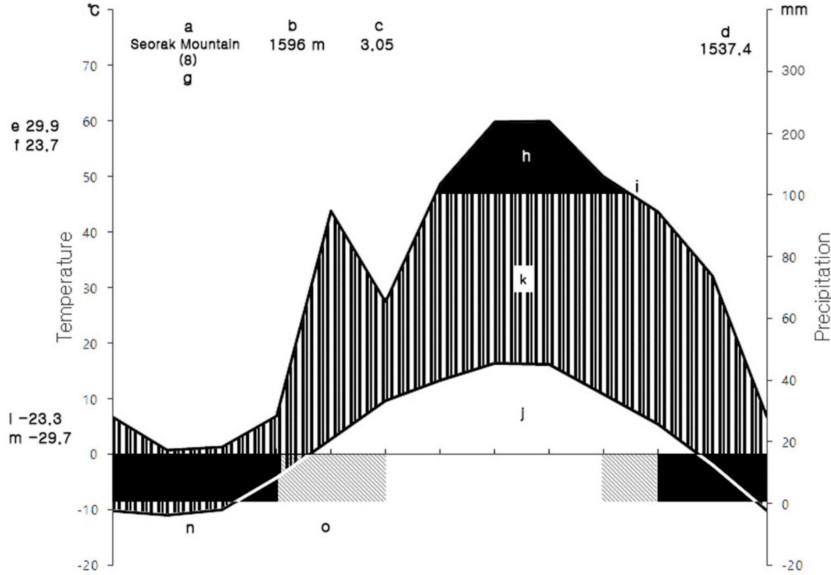

**Figure 1.** Climate diagram in Seorak Mountain for the recent eight years. Note: (**a**) Station name; (**b**) height above sea level (m); (**c**) mean annual temperature (°C); (**d**) mean annual amount of precipitation (mm); (**e**) absolute maximum temperature (°C); (**f**) mean daily maximum of the hottest month (°C); (**g**) number of years of observation; (**h**) average monthly precipitation exceeding 100 mm (black area); (**i**) monthly means of precipitation (mm); (**j**) monthly means of temperature (°C); (**k**) humid period (lined); (**l**) mean daily minimum temperature of the coldest month (°C); (**m**) absolute minimum temperature (°C); (**n**) months with a mean daily minimum temperature below 0 °C; and (**o**) months with an absolute minimum temperature below 0 °C.

*2.2. Survey and Data Collection*

Twenty-seven quadrats (maximum area, 400 m$^2$) were set up in Daecheongbong in 2016, based on vegetation canopy correlation and topography requirements. The study was carried out from March to October in 2016 and 2018 (Figure 2). The height and diameter at breast height (DBH) of all the species with a stem diameter of more than 2 cm were measured, and individual trees were labeled and numbered. The herbaceous layer was investigated to identify species present. Within each plot, the plant cover-abundance scale was recorded following the vegetation survey method of Braun-Blanquet [74]. To investigate the emergence and recruitment of *Abies nephrolepis* seedlings, all *Abies nephrolepis* seedlings of stem diameter less than 2 cm were analyzed, and the individuals were divided into five classes according to their height (< 10, 10–30, 30–50, 50–100, and > 100 cm).

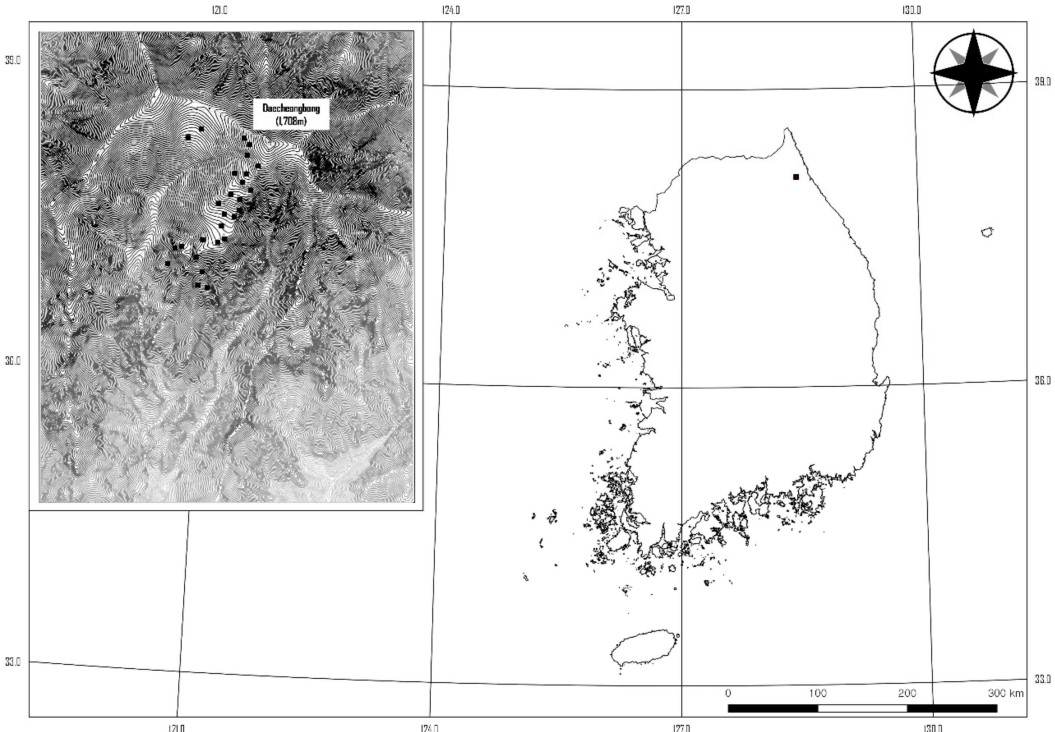

**Figure 2.** Location and topography of the study quadrats in Seorak Mountain, South Korea.

The environment factors were measured using the Garmin Montana 650 GPS equipment (Garmin, Olathe, KS, USA), including the latitude and longitude coordinates, aspect direction, and altitude above sea level. The slope degree was measured using a clinometer, within the four lines dividing a quadrat. For rocks of diameter 2 cm or more in the soil, five grades (< 5%, 5–15%, 16–30%, 31–50%, and > 51%) were used to record rock exposure above the soil surface. To measure canopy openness and transmittance, we used Fisheye-Nikkor 16-mm lens to photograph the center and four corners of the post.

### 2.3. Analysis Methods

The statistical program R 3.5.1 (The R Foundation) was used for data analysis. For DBH distribution, to verify whether there was a significant relationship between the independent variables, Chi-square test of independence was performed. To compare the changes in coniferous forest trees between 2016 and 2018, the number of trees per hectare was determined. To compare the changes in the number of *Abies nephrolepis* seedlings between 2016 and 2018, the number of seedlings per hectare was determined, and the data were transformed using the natural logarithm before analysis. For the total number of seedlings (seedling count), Shapiro–Wilk test was used, rejecting an alternative hypothesis if the seedling count was not normally distributed ($p = 0.74$). To assess whether there was a significant relationship between the independent variables and the distribution of seedling heights, Chi-square test of independence was conducted.

We report the means and standard error (SE; Table 1). The number of *Abies nephrolepis* seedlings, which was a dependent variable applied in the regression analysis, was replaced with natural logarithm. The abiotic factors used as independent variables were mean value rock exposure and canopy openness (Table 1). Geographical factors used as independent variables were mean altitude, aspect direction, and slope degree. Mean canopy openness was calculated using Gap light analyzer 3.0 based on the photographs captured using a Fisheye-Nikkor lens [75]. The biotic factors were the mean value of herbaceous cover, DBH, tree height, and species richness (Table 1).

**Table 1.** Description of the variables used in the R 3.5.1. The character of variables is denoted by A = abiotic, B = biotic; and the type by N = numeric. SE = standard error.

| Variable | Character | Type | Mean ± SE ($n = 27$) |
|---|---|---|---|
| Altitude (m) | A | N | 1532.1 ± 16.4 |
| Aspect direction (°) | A | N | 225.5 ± 11.9 |
| Slope degree (°) | A | N | 15.6 ± 1.6 |
| Rock exposure (%) | A | N | 32.4 ± 4.8 |
| Canopy openness (%) | A | N | 29 ± 3.3 |
| Species richness ($n$) | B | N | 28.7 ± 1.5 |
| Herbaceous cover (%) | B | N | 26.7 ± 1.5 |
| Basal area (m$^2$) | B | N | 0.034 ± 0.1 |
| Height (m) | B | N | 6 ± 0.7 |

We analyzed the data using simple and multiple linear regression, followed by F-tests. Prior to this, autocorrelation, homoscedasticity, multicollinearity, and outlier tests were performed to test the statistical assumptions. Stepwise backward elimination method was used to select independent variables for inclusion in the multiple regression [76,77]. Finally, the relative importance of the predictor variables was analyzed to explain the most important factors affecting the emergence of seedlings. For this, we used the standardized regression coefficients of predictor variables [78,79].

## 3. Results

### 3.1. Short-Term Dynamics of Tree Density and DBH Distribution in Conifer Forests

At the site, the distribution of DBH for *Pinus koraiensis* and *Quercus mongolica* was bell shaped, whereas that of *Abies nephrolepis*, *Betula ermanii*, and 15 taxa was inverse J shaped ($p < 0.001$; Figure 3).

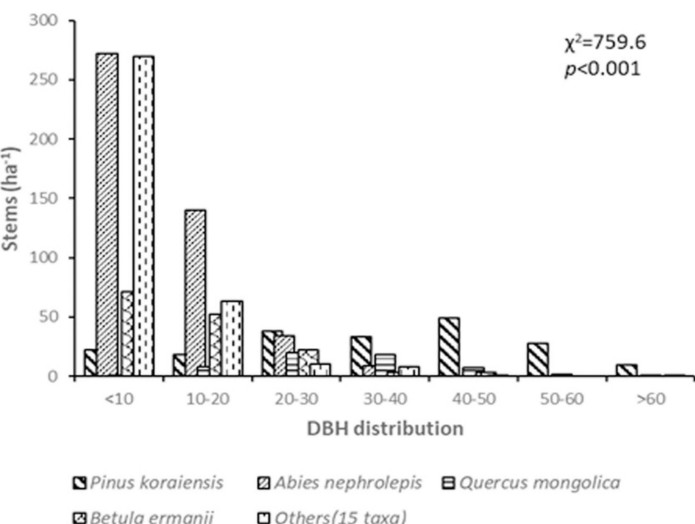

**Figure 3.** Woodys of diameter at breast height (DBH) distribution (data in 2018).

The population of trees > 2 cm in stem diameter decreased from 1173 ha$^{-1}$ in 2016 to 1151 ha$^{-1}$ in 2018 (Table 2). The change in the population of each species was as follows: *P. koraiensis* decreased from 201 to 198 ha$^{-1}$, *Abies nephrolepis* from 437 to 425 ha$^{-1}$, *Q. mongolica* from 60 to 59 ha$^{-1}$, and *B. ermanii* from 136 to 133 ha$^{-1}$. The number of dead trees was 27 ha$^{-1}$ over the two years. The number of recruited trees of *Abies nephrolepis* was 3 ha$^{-1}$, and that of other species was 2 ha$^{-1}$.

**Table 2.** Short-term dynamics of tree density in quadrat plots for two years (2016 and 2018).

| Species Population | Alive | | Dead | | *ND* | Recruited Stems |
|---|---|---|---|---|---|---|
| | 16′ | 18′ | 16′ | 18′ | | |
| *Pinus koraiensis* | 201 | 198 | 23 | 26 | 3 | 0 |
| *Abies nephrolepis* | 437 | 425 | 32 | 47 | 15 | 3 |
| *Quercus mongolica* | 60 | 59 | 3 | 4 | 1 | 0 |
| *Betula ermanii* | 136 | 133 | 5 | 8 | 3 | 0 |
| Dead tree (unknown) | - | - | 23 | 23 | - | 0 |
| Others (15 taxa) | 339 | 336 | 0 | 5 | 5 | 2 |
| Total | 1173 | 1151 | 86 | 113 | 27 | 5 |

*ND*: mortality of tree.

### 3.2. Short-Term Dynamics of Abies nephrolepis Seedlings

In all the quadrats, the number of *Abies nephrolepis* seedlings that emerged was 402 ha$^{-1}$ in 2016 and 528 ha$^{-1}$ in 2018 (Figure 4). The paired *t*-test analysis indicated that the change was significant from 2016 to 2018.

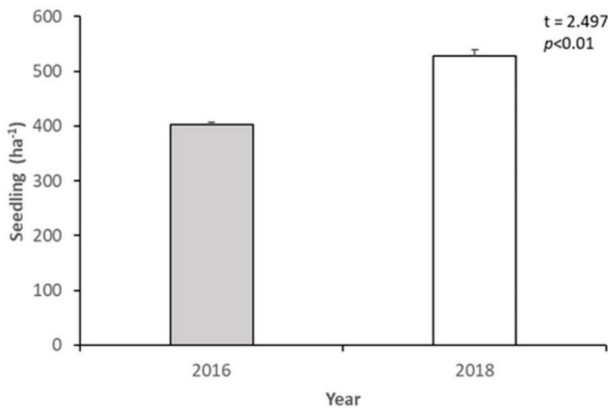

**Figure 4.** Short-term dynamics of *Abies nephrolepis* seedlings for two years (2016 and 2018).

The population dynamics of the species based on the height class data were as follows ($p < 0.001$; Figure 5): The number of seedings <10 cm in height increased from 108 ha$^{-1}$ in 2016 to 201 ha$^{-1}$ in 2018; seedings of 10–30 cm in height increased from 110 to 121 ha$^{-1}$; seedings of 30–50 cm in height increased from 46 to 99 ha$^{-1}$; seedings of 50–100 cm decreased from 61 to 50 ha$^{-1}$; and seedings of > 100 cm in height decreased from 77 to 57 ha$^{-1}$. As there was a forest gap in 2018, which was not there in 2016, the number of seedlings < 50 cm in height was higher in 2018, but seedlings > 50 cm decreased due to a failure of ingrowth.

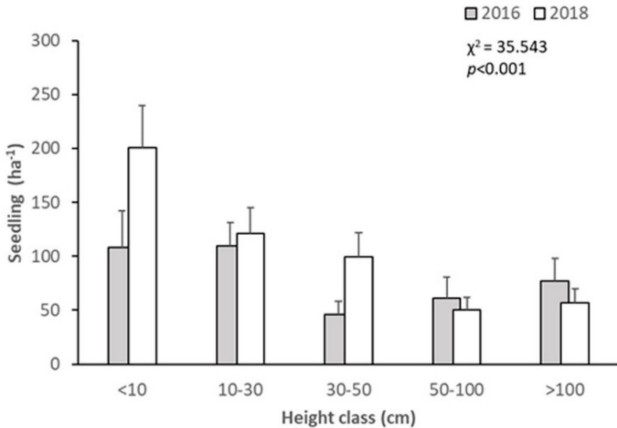

**Figure 5.** Short-term dynamics of *Abies nephrolepis* seedlings along height class for two years (2016 and 2018).

### 3.3. *Abies nephrolepis Seedlings and Their Relationships with Environmental Factors*

The simple linear regression of the density of *Abies nephrolepis* seedlings and environmental factors showed that altitude, slope degree, species abundance, and height presented low coefficient values, and they were not statistically significant (Figure 6). The mean ± SE of the total number of *Abies nephrolepis* seedlings were 285.5 ± 68.7 for the 27 study sites. In contrast, aspect direction, rock exposure, canopy openness, herbaceous cover, and DBH presented high coefficient values, and they were statistically significant.

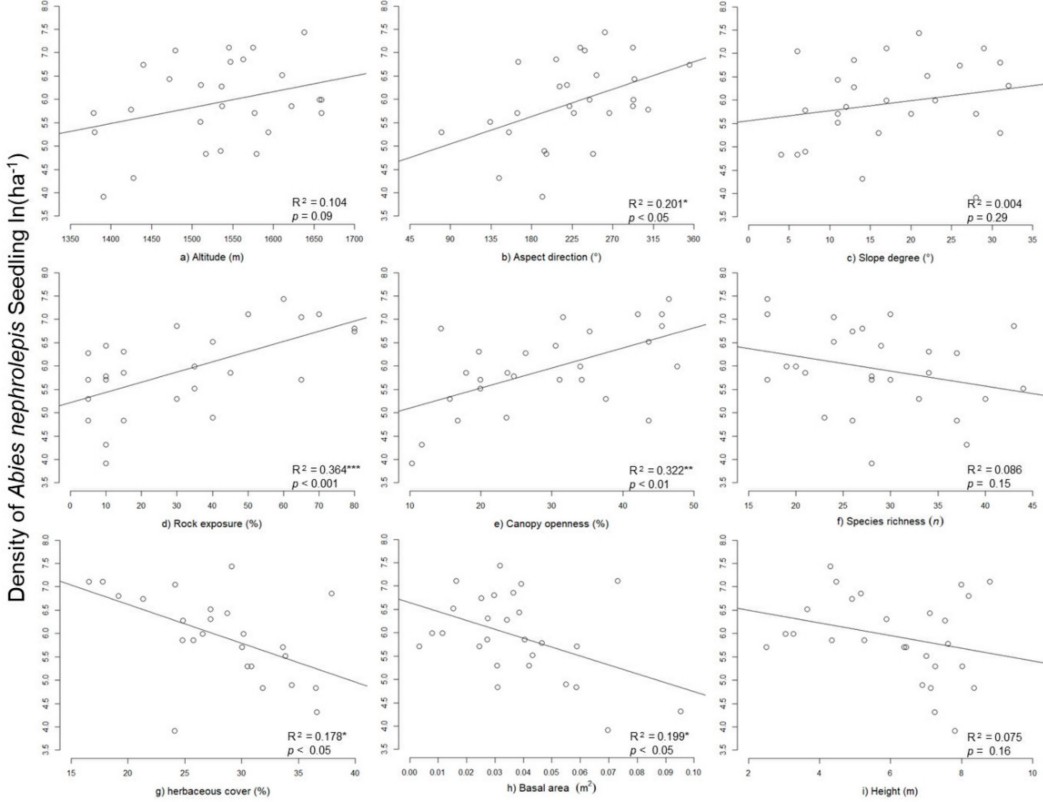

**Figure 6.** Relationships between seedlings and variables by simple linear regression analysis. Field data from 27 plots, investigated in Daecheongbong area in 2018, are represented by white points. (Note. Significance levels: * $p < 0.05$; ** $p < 0.01$; *** $p < 0.001$.)

Among the variables selected using the stepwise backward elimination method, rock exposure, canopy openness, and species richness presented positive relationships, whereas herbaceous cover and basal area presented a negative relationship (Table 3). Although basal area was not a significant variable ($p = 0.1149$), it was not removed by the stepwise regression method. This is because the other variables explained a relatively high proportion of the variance.

**Table 3.** Results of the multiple regression model for *Abies nephrolepis* seedling.

| Variable | Estimate | SE | *t*-Value | *p*-Value |
|---|---|---|---|---|
| (Intercept) | 4.3783 | 0.7162 | 6.113 | 0 *** |
| Rock exposure (%) | 0.0202 | 0.0048 | 4.221 | 0.0004 *** |
| Canopy openness (%) | 0.0345 | 0.0109 | 3.167 | 0.0046 ** |
| Species richness (n) | 0.0442 | 0.0184 | 2.395 | 0.0260 * |
| Herbaceous cover (%) | −0.0368 | 0.0155 | −2.380 | 0.0269 * |
| Basal area (m$^2$) | −10.6060 | 6.4488 | −1.645 | 0.1149 |

Coefficient of determination, $R^2 = 0.701$; adjusted $R^2 = 0.630$; the degree of freedom F-statistic, 9.867; and statistical probability of multiple regression, $p = 0.0000578$. (Note. Significance levels: * $p < 0.05$; ** $p < 0.01$; *** $p < 0.001$.)

The relative importance calculated using the standard regression coefficient of determination ($R^2$) revealed that basal area had greater explanatory power than herbaceous cover and species richness (Figure 7).

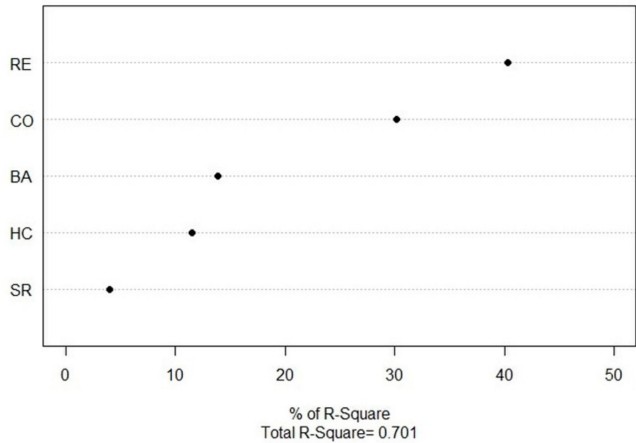

**Figure 7.** The selection variables by backward elimination are about multiple regression analysis (Table 3). The relative importance of predictor variables is represented by black points. (Note. RE: Rock exposure; CO: Canopy openness, BA: Basal area; HC: Herbaceous cover; SR: Species richness).

## 4. Discussion

According to the DBH distribution, the largest number of individuals had DBH <10 cm. Many seedlings were in the emergence stage of growth, a strategy allowing survival and growth after various disturbances, such as the strong winds, typhoons, and long winters that occur in the area. Therefore, we infer that sustainable mast seeding promoted seedling density, and that some species were more tolerant of the extreme conditions than others. Relating our findings to the climate of the area (Figure 1), we infer that April precipitation partially ends the spring drought, and that the high precipitation from June to September greatly influences seedling survival. It appears that the low temperature that continues into May does not seem to constrain growth and survival [80]. Ultimately, we consider *Abies nephrolepis* in this area to exemplify the pollination efficiency theory.

Forests that have a bell-shaped DBH distribution lack old and young trees [81–83]. Therefore, such forests are unsustainable and reflect population decline caused by external disturbances [84,85]. An inverse J-shaped DBH distribution appears because of high initial mortality of young trees [85]. Old-growth and virgin forests generally show a typical inverse J-shaped DBH distribution [86–89].

As the inverse J-shaped DBH distribution is observed for trees at all ontogenetic stages, it could be a structure that is relatively more stable than the bell-shaped DBH distribution [90]. The total number of *Abies nephrolepis* seedlings on Seorak Mountain was higher than that of *Abies koreana* seedlings on Jiri Mountain [63]. This indicates that the regeneration of seedlings differs in mixed communities co-dominated by *B. ermanii*, or in pure communities of fir species. Moreover, in the pure communities, seedling regeneration progresses rapidly, whereas in the mixed communities it progresses slowly due to suppression by competition [91]. These differences are due to the difference in the growth rate and shade tolerance among the fir species [36,91,92]. There was no ingrowth of *Abies koreana* seedling on Jiri Mountain [63]. Therefore, the growth of seedlings varies with canopy layer [93]. The growth and morphological changes of seedlings of *Abies* spp. and *Picea* spp. were more sensitive to light intensity than those of *Pinus* spp. [35]. In addition, seedlings cannot grow in alpine areas in winter due to the layer of debris, water supply shortage, and snowfall, causing in high seedling mortality [29,92,94].

The linear relationship revealed that as aspect direction approached 225–270°, the number of seedlings that emerge increased ($R^2 = 0.201$, $p < 0.05$), indicating that photoperiod and solar radiation affect seedling emergence via aspect direction [95,96]. Seedling emergence differed with rock exposure ($R^2 = 0.364$, $p < 0.001$). In the coniferous forests, micro-topography due to rock exposure mostly results in moss coverage, and the survival and mortality of seedlings vary with moss coverage area [37,38,40]. Because moss species function as sponges, they have a high ability to absorb water, effectively absorbing and fixing nutrients [27,97,98]. In addition, at Seorak Mountain, *Abies nephrolepis* seedlings may emerge or survive on the lee side of rocks, thus receiving protection from wind damage.

Seedling emergence increased with canopy openness ($R^2 = 0.322$, $p < 0.01$). The growth of species in the understory is influenced by transmittance, solar radiation, and moisture, which are affected by canopy openness [40,96,97]. Herbaceous cover was at 15–30%, and the number of seedlings tended to decrease sharply as herbaceous cover increased ($R^2 = 0.268$, $p < 0.01$). With respect to the survival of tree seedlings, it has been reported that Cyperaceae and Gramineae species in the herbaceous layer assist by maintaining the water content of the soil [39]. As Bryophyta, and Gramineae and Cyperaceae species coexist in the coniferous forests in this area, the survival of seedlings is positively influenced, but structural deterioration due to the invasion of the vine species is a concern that should be addressed [21,57]. We found that the lower basal area of trees at our study site was correlated with the emergence of more seedlings ($R^2 = 0.199$, $p < 0.05$), because of the co-dominance of *P. koreana*, which has a large basal area, in the canopy layer [21,55,56]. This suggests that the basal area is mostly covered by small woody-type species, which is similar to the results observed elsewhere in the small woody-type *Abies nephrolepis* [51].

Our findings indicate that the various environmental factors mentioned above affect the generation of *Abies nephrolepis* seedlings. Seorak Mountain in Korea has more rocky ground than woody debris, and the forest has appropriate canopy openness. Therefore, we considered that the rock provides wind protection and a substrate for moss, which may increase the density of *Abies nephrolepis*. In addition, the canopy openness has an overall synergistic effect. Therefore, these environmental factors have a positive effect on the generation and survival of *Abies nephrolepis* seedlings. In contrast, the herbaceous layer has a negative effect on *Abies nephrolepis* seedlings. Since Seorak Mountain does not have a gentle topography, it is difficult to develop microsite grassland community on the slopes. The slope may explain why the herbaceous layer comprises mostly vine plants and genera, rather than grass species. Regarding co-dominance or tree on Seorak Mountain, most of the *Abies nephrolepis* trees negatively affect the total basal area, because of their small DBH, in contrast to the large DBH of *P. koreana*. This shows that analysis of the DBH distributions within a habitat can help to elucidate the relationships between environmental factors and vegetation density and occurrence. In summary, Seorak Mountain has a variety of habitats with suitable rock exposure and canopy openness, which affect the forest regeneration process.



## 5. Conclusions

Our results reveal that the seedlings' density was associated with environmental variables. The results are consistent with existing qualitative data. They may also be relevant for the in-situ and ex-situ conservation of subalpine conifers. In addition, it is important to incorporate multiple variables of seedling emergence into predictions of subalpine forest dynamics. The structure of the coniferous forest was considered sustainable for the subsequent regeneration, because *Abies nephrolepis* was present at all ontogenetic stages. The higher number of dead trees of this species reflects the fact that *Abies nephrolepis* population density was higher than that of other species. There is an ecological difference between *Abies koreana* and *Abies nephrolepis* in terms of emergence and survival. Therefore, comparative analysis of *Abies nephrolepis* and *Abies koreana* is required to elucidate the ecological differences in seed activity, tolerance, and growth. We found that *Abies nephrolepis* seedlings were affected by canopy openness and herbaceous cover, which is with the view that seedling growth is related to light intensity, which depends on the upper story.

To the best of our knowledge, we present the first description of the effects of environmental factors on *A. nephrolepis* seedling density in subalpine forests. This study was carried out to analyze associations between biotic and abiotic factors and *Abies nephrolepis* seedlings' density on Seorak Mountain. Further investigation of climatic factors is necessary to understand the overall ecological divers of seedling density. In addition, a comparative analysis of *Abies holophylla* and *Abies koreana* seedlings, including *Abies nephrolepis* seedlings, might be a useful approach to understand the diverse ecological characteristics of the fir genus. Our findings in terms of mast seeding or masting theory are limited, because we have investigated only two years of data. Therefore, we intend to carry out continuous monitoring in order to further understand and elucidate seedling dynamics, survival, and growth of *Abies nephrolepis* in this ecosystem.

**Author Contributions:** Formal analysis, J.K.; investigation, J.K. and C.Y.; methodology, J.K.; project administration, J.-H.L.; supervision, C.Y.; validation, C.Y.; writing–original draft, J.K.; writing–review and editing, J.-H.L.

**Funding:** This research received no external funding.

**Acknowledgments:** In this study, Forestry and Forestry Sector Climate Change Impact Vulnerability Assessment and Adaptation Research Project in National Institute of Forest Science (Commissioned Research Project 'Investigation and Analysis of Changes on Subalpine Conifer Forests in Seorak mountain') were carried out as part of long-term ecological monitoring.

**Conflicts of Interest:** The authors declare no conflicts of interest.

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
