# Peer review of "Dynamics of Abies nephrolepis Seedlings in Relation to Environmental Factors in Seorak Mountain, South Korea"

_forests, doi:10.3390/f10080702_

Round 1
Reviewer 1 Report
Here the authors describe seedling dynamics for Abies nephrolepsis over a period of two years. They found a general decrease in seedling stem density over this period, and also
found that seedling density was rather strongly driven by rock exposure and canopy openness. There were also some other predictor variables that had pretty decent correlations with seedling density. These results are quite interesting, as many times these types of enviornmental correlations with seedling density turn out weak, even when you expect them to be strong. At the same time, the very short duration of the study makes it quite difficult to make any general conclusions or predictions. Seedling dynamics are highly variable and stochastic, and seedling density, in particular, can change dramatically from year to year. A two-year census interval makes it quite difficult to quantify this variability. As such, it is difficult to determine whether the models the authors develop here would be very useful across different years.
While I think descriptive studies such as this one are important, the presentation of the manuscript is extremely general and vague. The authors present no real research questions, hypotheses, rationales or goals for the study, etc. As such, it is very difficult to see how readers would find this study to be useful outside of the specific forest and species that the authors use here. I think the underlying field data and methodologies are sound, but the presentation of the manuscript needs a huge amount of improvement.
Likewise, the writing certainly needs to be greatly improved. It is by no means bad, but there are many obvious grammatical and stylistic errors throughout that an English editor would be able to correct easily. I think the authors should consider doing this. However, Tthere are also more major problems, for example with sentences that are very unclear, and the overall organization of the manuscript needs to be revised as well. This may take a large amount of work to improve.
Line Comments:
L14: You bring up a different species here than the one you say above that you are studying. This is very confusing. And what is the purpose of making this comparison between the two species in terms of their diameter distributions?
L21: Among each other, or with seedling density? This is not at all clear.
L23: Here too, a relationship with what? It is not clear what relationships you are describing here.
L25: In what way? You do not explain what the response variable for the seedlings is that you are modelling. I assume you mean density, but this could also be growth or survival, etc. No way to tell based on the absrtact alone.
L27: This is an extremely general sentence for the second to last sentence of your abstract. Remove this. It is just a waste of space.
L27: Your results do not show a causal relationship, because this is just an observational study. Assigning causation from a regression model based on observational data is not possible.
L28: Now you mention 'natural emergence,' which you have not mentioned before. This could imply that you are modelling recruitment. Is that true. As mentioned above, we have no way of undertanding what the response variable in your model is. The abstract needs a lot of work to make it more clear.
L33: An ecocline invovles a gradient in species composition, right? So it makes no sense to say that the forest is 'subjected to' an ecocline. The forest is part of the ecocline, right? Perhaps you mean an extreme gradient in climate or soils?
L37: Not very clear what you mean by 'lack of understanding.' These systems have been studied extensively. not sure you can make this claim, especially without including some sort of citation for this statement.
L38: This last sentence implies that your own work will be such a study. However, you just have two years of data for a single species, which is hardly 'extensive.'
L42: This is not true. It is important for many forest studies, but for some it might be meaningless. DBH is just a simple way of measuring tree size. No need to state here whether it is important or not. Also, it is not appropraite to provide an overview of DBH at the beginning of your introduction section.
L43: What do you mean by 'which constitutes forest canopy?' How does DBH consitute the canopy? Very unclear.
L45: time difference in what? And what is the significance of 5 and 10 years? This entire paragraph seems completely unnecessary. There is no need to explain DBH and long-term forest monitoring here. Readers of this journal will be very familiar with long-term forest monitoring studies.
L46: Rationale for understory studies is not clear here. It is obvious that seedlings can be monitored in a long-term study. I do not understand what point you are trying to make here.
L48: "Particularly, in the understory, the dynamics of seedlings, important development stage, can help determine whether a plant can survive in a habitat." This sentence does not make any sense.
L50: Much of the information in this paragraph and the next would be more appropriate in the methods section. Also, towards the end of your introduction it is very apparent that there are really no research motivations for this work. There are no hypotheses, questions, etc. presented that show the reader why you decided to conduct this work. this definitely needs to be included.
L119: You already knew this though, so this is not really a result.
L120: This is the first time you mention this oak species. Until now, you have only mentioned the two conifers, and have made it sound like those two species dominate the canopy. This is confusing to bring up this new species here.
L156: Seedling what? It is apparent from the axes that this is density, but you need to clearly state this in the figure caption. This was a big problem in the abstract as well. Also, you need to include p-values on these panels.
Author Response
Response to Reviewer 1 Comments
Point 1: Here the authors describe seedling dynamics for Abies nephrolepsis over a period of two years. They found a general decrease in seedling stem density over this period, and also found that seedling density was rather strongly driven by rock exposure and canopy openness. There were also some other predictor variables that had pretty decent correlations with seedling density. These results are quite interesting, as many times these types of enviornmental correlations with seedling density turn out weak, even when you expect them to be strong. At the same time, the very short duration of the study makes it quite difficult to make any general conclusions or predictions. Seedling dynamics are highly variable and stochastic, and seedling density, in particular, can change dramatically from year to year. A two-year census interval makes it quite difficult to quantify this variability. As such, it is difficult to determine whether the models the authors develop here would be very useful across different years. While I think descriptive studies such as this one are important, the presentation of the manuscript is extremely general and vague. The authors present no real research questions, hypotheses, rationales or goals for the study, etc. As such, it is very difficult to see how readers would find this study to be useful outside of the specific forest and species that the authors use here. I think the underlying field data and methodologies are sound, but the presentation of the manuscript needs a huge amount of improvement. Likewise, the writing certainly needs to be greatly improved. It is by no means bad, but there are many obvious grammatical and stylistic errors throughout that an English editor would be able to correct easily. I think the authors should consider doing this. However, There are also more major problems, for example with sentences that are very unclear, and the overall organization of the manuscript needs to be revised as well. This may take a large amount of work to improve.
Response: These ecosystems are in a constant state of flux where the only constant is change, and that change can be linear or nonlinear. Therefore, we are continually monitoring the region, and the monitoring will be occurring at 2-year intervals. Accordingly, the first 2 years of this project corresponds to a time where there is still a fuzzy concept about monitoring. The study sites are sub-alpine coniferous forests, which are not yet widely known in East Asia. For this long-term monitoring project, we chose one species to observe the seedling dynamics over a short period of time.
In addition to clarifying this general context and goal of our work, we have inserted new text in the abstract of this revised manuscript. The original version of the manuscript was edited by a professional English editing service before submission. The same service has reviewed the revised manuscript.
Line Comments:
Point 2: L14: You bring up a different species here than the one you say above that you are studying. This is very confusing. And what is the purpose of making this comparison between the two species in terms of their diameter distributions?
Response: This analysis was performed to explain differences of the population structure through comparison of the diameter distribution in the subalpine forest in Seorak Mountain, Korea. In addition, by explaining the community structure, we can gain a rough understanding of the transition progress and indicator species in a habitat. Previous studies have generally described the community structure of a corresponding plot (e.g., Liu et al., 2014; Omelko et al., 2016).
Point 3: L21: Among each other, or with seedling density? This is not at all clear.
Response: We revised the expression per the reviewer’s suggestion for clarity to: “… show significant linear relationships with the density of A. nephrolepis seedlings.”
Point 4: L23: Here too, a relationship with what? It is not clear what relationships you are describing here.
Response: We added the following sentence to clarify this relationship: “A multiple linear regression analysis on the density of A. nephrolepis seedlings revealed …”
Point 5: L25: In what way? You do not explain what the response variable for the seedlings is that you are modelling. I assume you mean density, but this could also be growth or survival, etc. No way to tell based on the absrtact alone.
Response: We clarified this analysis as follows: “Overall, the A. nephrolepis seedling density was most strongly influenced by the presence of large rocks; the rocks provide shelter from harsh winds and a substrate for Bryophta to grow on.”
Point 6: L27: This is an extremely general sentence for the second to last sentence of your abstract. Remove this. It is just a waste of space.
Response: We removed the original sentence according to the reviewer's suggestion and added the following “Moreover, we considered that appropriate canopy openness creates a synergistic relationship.”
Point 7: L27: Your results do not show a causal relationship, because this is just an observational study. Assigning causation from a regression model based on observational data is not possible.
Response: We agree with this suggestion. We carefully interpreted these results about correlation relationship. We have added text to provide relevant grounding theories and hypotheses in the Introduction to support our conclusion as follows: “In particular, in subalpine and alpine zones, the regeneration of tree seedlings depends on various environmental factors. The climate at high elevation is characterized by strong seasonal variation with short, moderately warm, and moist summers, and long, extremely cold, and dry winters [25-28]. In general, alpine plants adapted to dry conditions can use water very [29-31]. Wind conditions have the greatest influence on physiological change and damage of plants at the treeline; rocks can provide shelter from the wind for tree seedlings during the germination and early growth stages [32-35]. In addition, the influence of canopy gaps on forest processes such as stand structure, regeneration, and dynamics of the understory should be considered [36-38]. These various physical factors contribute to some extreme environments. Moreover, analysis of the dynamics of seedlings with manipulations of neighboring herbaceous cover and microclimate has revealed that protection from bright sunlight, low temperature, and water stress are also particularly important factors that facilitate growth and survival [39-42].”
Point 8: L28: Now you mention 'natural emergence,' which you have not mentioned before. This could imply that you are modelling recruitment. Is that true. As mentioned above, we have no way of undertanding what the response variable in your model is. The abstract needs a lot of work to make it more clear.
Response: We have now tried to clarify that seedling density was the response throughout. As the reviewer suggested, we added the following sentence to clarify the conclusion: “Overall, the A. nephrolepis seedling density was most strongly influenced by the presence of large rocks; the rocks provide shelter from harsh winds and a substrate for Bryophta to grow on. Moreover, we considered that appropriate canopy openness creates a synergistic relationship. We found a positive correlation between the seedling density of subalpine forests and certain physical environmental factors. These results show that generally known ecological hypotheses on the roles of rocks and canopy are true even in this habitat.”
Point 9: L33: An ecocline invovles a gradient in species composition, right? So it makes no sense to say that the forest is 'subjected to' an ecocline. The forest is part of the ecocline, right? Perhaps you mean an extreme gradient in climate or soils?
Response: As the reviewer suggested, we changed "subjected to" to "a part of" in this sentence.
Point 10: L37: Not very clear what you mean by 'lack of understanding.' These systems have been studied extensively. not sure you can make this claim, especially without including some sort of citation for this statement.
Response: As the reviewer suggested, we added the following sentence with relevant citations to clarify the background related to our aims: “… species at high elevations are more sensitive to climate change than those in lowland areas [7-9], because the long lifespan of trees or narrow-range species does not allow for rapid adaptation to environmental changes [10].”
The intention of the original statement "lack of understanding" was that there is still a lack of monitoring alpine and sub-alpine areas in other regions. The sentence has been rephrased to more clearly express our point. “Therefore, there is a need for adopting various approaches and interpretations about these ecosystems.”
Point 11: L38: This last sentence implies that your own work will be such a study. However, you just have two years of data for a single species, which is hardly 'extensive.'
Response: As the reviewer suggested, we added the following sentence to clarify our position and aims: “Therefore, there is a need for adopting various approaches and interpretations about these ecosystems.”
This last sentence also indicates that extensive research is needed in this field on an ongoing basis. Of course, this is meant to support our current research and short- and long-term aims. In addition, since the data for 2 years is part of various studies, we consider it to be included in the general idea that adopting various approaches and interpretations is an "extensive" research approach.
Point 12: L42: This is not true. It is important for many forest studies, but for some it might be meaningless. DBH is just a simple way of measuring tree size. No need to state here whether it is important or not. Also, it is not appropraite to provide an overview of DBH at the beginning of your introduction section.
Response: We appreciate the reviewer's perspective, but we appear to have a slight difference in opinion on the importance of DBH. Although DBH is a simple method, it is also considered to be the most powerful and quickest way to collect information in a stand. The reason is that there are many basic forest monitoring projects that focus on DBH. However, in these studies, understory dynamics are excluded or not largely considered in the forest monitoring. Therefore, this statement refers to the paradoxical aspect of long-term monitoring of diameter growth.
Point 13: L43: What do you mean by 'which constitutes forest canopy?' How does DBH consitute the canopy? Very unclear.
Response: As the reviewer suggested, we added the following sentence to clarify: “… of the dominant trees that make up the forest canopy [11].”
Point 14: L45: time difference in what? And what is the significance of 5 and 10 years? This entire paragraph seems completely unnecessary. There is no need to explain DBH and long-term forest monitoring here. Readers of this journal will be very familiar with long-term forest monitoring studies.
Response: In general, we agree with this suggestion. However, our research is based on the limitations of "time" in long-term forest monitoring. In recent years, most long-term monitoring studies have observed responses over a long period of time (such as interval 5 and 10 years or longer in some cases). However, this may be a rare study. Thus, we have been studying what we are missing during that "time". We are particularly interested in the dynamics of seedlings and plan to extend this study to suggest various ecological changes and related environmental factors in subalpine forests.
Point 15: L46: Rationale for understory studies is not clear here. It is obvious that seedlings can be monitored in a long-term study. I do not understand what point you are trying to make here.
Response: Again, in a general sense, we would agree. This point is based on the reviewer's broad knowledge, and most of the previous long-term studies reported more than a few years of data. However, this suggests that many short-term studies on the "understory" have been neglected, which raises several questions: how do the seedling dynamics change in a short period of time? What if this happens in a specific area or habitat? What is the relationship between the environmental variables and seedling density? Our study was built on these questions. Therefore, our research is not focused on long-term data, but rather on the "understory" that shows dramatic dynamics by simply observing changes in seedlings for 2 years.
Point 16: L48: "Particularly, in the understory, the dynamics of seedlings, important development stage, can help determine whether a plant can survive in a habitat." This sentence does not make any sense.
Response: This sentence was meant to emphasize the relationship between a habitat and understory, and regeneration of the seedlings. To reinforce the Introduction section, this sentence was moved up and incorporated into another paragraph, and revised for clarity as indicated above in the response to comment #6.
Point 17: L50: Much of the information in this paragraph and the next would be more appropriate in the methods section. Also, towards the end of your introduction it is very apparent that there are really no research motivations for this work. There are no hypotheses, questions, etc. presented that show the reader why you decided to conduct this work. this definitely needs to be included.
Response: In the previous revised manuscript, we stated our specific hypothesis and have now added more detail to clarify our specific goals. We revised this text per the reviewer’s suggestion, and some paragraphs were moved to the Methods section as appropriate.
Point 18: L119: You already knew this though, so this is not really a result.
Response: For the forest type, we state that “P. koraiensis and A. nephrolepis were co-dominant in most of the coniferous forests in the study site. However, the diameter distribution of the two species was different.”
Point 19: L120: This is the first time you mention this oak species. Until now, you have only mentioned the two conifers, and have made it sound like those two species dominate the canopy. This is confusing to bring up this new species here.
Response: Our research is part of a long-term monitoring project of community structure dynamics. Oak is one of the constituent species forming the community structure in the sub-alpine forest. In fact, two species of conifers dominate the canopy, and other related papers have also mentioned other major species(e.g., Kuulauvainen et al., 1996; Motta et al., 1999; Manabe et al., 2000; Schwartz et al., 2005; Penuelas et al., 2007; Lamedica et al., 2011). Therefore, we have also mentioned two major conifer species in addition to other species that are present in Seorak Mountain.
Point 20: L156: Seedling what? It is apparent from the axes that this is density, but you need to clearly state this in the figure caption. This was a big problem in the abstract as well. Also, you need to include p-values on these panels.
Response: As the reviewer suggested, we have clarified that seedling density is measured as the response variable, and have also included p-values in the figure panels.

Reviewer 2 Report
I have now reviewed the manuscript by Ji-Dong Kim. The research is interesting and the work done is valuable with very interesting data on the topic. However, some concerns need to be addresses prior to publication. Please see below:
- Abstract: these are the number in of trees and seedlings by ha obtained from 27 plots. Thus, you mean the total number of trees or this is about a mean from plots? In this case, information about mean values and the dispersion of the mean (i.e. standard error, standard deviation) is needed in order to make comparisons among species and so on.
- Introduction: The introduction need to be modified adding more information about the addressed topics. Information about the factors directly or indirectly affecting natural regeneration (in which way and at which extent environmental factors may affect natural regeneration on the particular case of alpine forest). The objectives need to be clearly stated. As the author stated (line 61-64 ) The main objective is to investigate recruitment dynamics of a particular specie. Why did you study more than one specie (see figure 2)? You can have a wide scope adding more target species to the objectives. For the second objective, you have to be more concise, which environmental factors are you talking about? All these factor should be addressed on the introduction section.
- Material and methods: Please add an information about the forest type, species, and management and so on. All this date will help to know better the study and forest area.
L72: How did you surveyed herbaceous and vegetation identity, sociability and dominance. All the methodology need to be clearly exposed.
Results: Along the results, readers need to know the mean values and standard deviation in order to know whether statistical differences exit comparing species and experimental factors. On the other hand, the R squares of the linear regressions (fig 5) are a bit low to be considered as explicative enough. Did you explore different to linear regression relationships? For example, log regression and so on.
- Discussion: As in the introduction section, I think the author should go deep with the explanations about how environmental factor may affect/alter natural regeneration of the target species. It is not about to say grass cover avoid seedling development, it is also about to say why? When? How? Grass cover may alter initial recruitment of target species in Alpine forest and secondly, try to relate your data with international studies on the same topic.
I hope my comments help to improve the manuscript.
Author Response
Response to Reviewer 2 Comments
I have now reviewed the manuscript by Ji-Dong Kim. The research is interesting and the work done is valuable with very interesting data on the topic. However, some concerns need to be addresses prior to publication. Please see below:
Thank you for review.
Point 1: - Abstract: these are the number in of trees and seedlings by ha obtained from 27 plots. Thus, you mean the total number of trees or this is about a mean from plots? In this case, information about mean values and the dispersion of the mean (i.e. standard error, standard deviation) is needed in order to make comparisons among species and so on.
Response: We apologize for the confusion caused by neglecting to clarify these details. We have now clarified that the data represent “The total number of” as relevant throughout the manuscript.
Point 2: - Introduction: The introduction need to be modified adding more information about the addressed topics. Information about the factors directly or indirectly affecting natural regeneration (in which way and at which extent environmental factors may affect natural regeneration on the particular case of alpine forest). The objectives need to be clearly stated. As the author stated (line 61-64 ) The main objective is to investigate recruitment dynamics of a particular specie. Why did you study more than one specie (see figure 2)? You can have a wide scope adding more target species to the objectives. For the second objective, you have to be more concise, which environmental factors are you talking about? All these factor should be addressed on the introduction section.
Response: We revised the Introduction per the reviewer’s apt suggestions, including the addition of the following paragraph: “In particular, in subalpine and alpine zones, the regeneration of tree seedlings depends on various environmental factors. The climate at high elevation is characterized by strong seasonal variation with short, moderately warm, and moist summers, and long, extremely cold, and dry winters [25-28]. In general, alpine plants adapted to dry conditions can use water very [29-31]. Wind conditions have the greatest influence on physiological change and damage of plants at the treeline; rocks can provide shelter from the wind for tree seedlings during the germination and early growth stages [32-35]. In addition, the influence of canopy gaps on forest processes such as stand structure, regeneration, and dynamics of the understory should be considered [36-38]. These various physical factors contribute to some extreme environments. Moreover, analysis of the dynamics of seedlings with manipulations of neighboring herbaceous cover and microclimate has revealed that protection from bright sunlight, low temperature, and water stress are also particularly important factors that facilitate growth and survival [39-42].”
In Seorak Mountain, A. nephrolepis has the most clear regeneration pattern compared to that of other species. A. nephrolepis seedlings occur most frequently, and it is difficult to confirm the occurrence of seedlings of other species. We have been measuring the density of seedlings for other species as well; however, it is difficult to interpret these results clearly and there are many aspects that make it difficult to reliably apply statistical analysis. We have also clearly stated our second goal in the manuscript.
Point 3: - Material and methods: Please add an information about the forest type, species, and management and so on. All this date will help to know better the study and forest area.
Response: We have added more details of the study site per the reviewer’s suggestion to the Material and Methods as follows: “The vegetation of Seorak Mountain comprises mixed forest or pure forest of evergreen conifers such as Pinus densiflora, Abies holophylla, Pinus koraiensis, and Abies nephrolepis. In addition, shrub forests such as Lonicera caerulea, Crataegus komarovii, Taxus caespitosa, and Pinus pumila are perched at the top of the mountain, and alpine plants such as Anemone narcissiflora, Geranium eriostemon, and Arctous ruber are distributed throughout the area.”
Point 4: L72: How did you surveyed herbaceous and vegetation identity, sociability and dominance. All the methodology need to be clearly exposed.
Response: We provided the following details on the methodology to the revised manuscript per the reviewer's suggestion: “The herbaceous layer was investigated to identify the species. Within each plot, the plant cover-abundance scale was recorded following the vegetation survey method of Braun-Blanquet (1965) [70].”
Point 5: Results: Along the results, readers need to know the mean values and standard deviation in order to know whether statistical differences exit comparing species and experimental factors. On the other hand, the R squares of the linear regressions (fig 5) are a bit low to be considered as explicative enough. Did you explore different to linear regression relationships? For example, log regression and so on.
Response: We altered the expression per the reviewer’s suggestion. We added several relevant sentences: “The mean (±SE) of the total number of A. nephrolepis seedlings was 285.5±68.7 for the 27 study sites.”
In addition to linear regression analysis, other statistical treatments have been considered. However, since our seedling data is a survey of the frequency of the individual; binomial or polynomial regression is not appropriate.
Point 6: - Discussion: As in the introduction section, I think the author should go deep with the explanations about how environmental factor may affect/alter natural regeneration of the target species. It is not about to say grass cover avoid seedling development, it is also about to say why? When? How? Grass cover may alter initial recruitment of target species in Alpine forest and secondly, try to relate your data with international studies on the same topic.
Response: We have revised the Discussion and added more explanation on the influence of environmental factors to the Discussion per the reviewer's suggestion as follows: “In contrast, the herbaceous layer has a negative effect on A. nephrolepis seedlings. Since Seorak Mountain does not have a gentle topography, it is difficult to develop a grassland community at a microsite. For this reason, the herbaceous layer could be considered to have a cover-abundance scale of vine plants or other competing plants rather than grass species.”
I hope my comments help to improve the manuscript.

Round 2
Reviewer 1 Report
The authors of this manuscript have made many important and significant improvements to their manuscript. At the same time, my previous review of this work indicated that very significant improvements in the writing and focus of the manuscript would be necessary for this to be considered more seriously for publication. While I think the authors have indeed made important steps in the right direction, I still do not find the manuscript to be suitable for consideration at Forests.
One of my most important suggestions was that the authors should add clear research questions/hypotheses, and also provide further details on the motivations behind their work. Unfortunately, the authors seem to have ignored this comment altogether. As a result, the manuscript still severely lacks focus and organization. This is evident throughout the entire text, from introduction through conclusions. If there were two or three clear research questions presented at the end of the introduction, then the entire rest of the manuscript could be organized around these questions. This would admittedly be a lot of work, but I do find this to be necessary. As it stands, the work still seems highly descriptive and extremely focused on the single study site that the authors collected data from. As a result, it is not at all clear what value this work would have to the general readership of Forests.
While the authors did a good job at responding to my comments at length, I often found their responses to either be confusing, or inadequate for the most part. As one example, I asked the authors to explain why they bring up Pinus koraiensis in the abstract and elsewhere, when the title and beginning of abstract make it clear that the study focuses on Pinus nephrolepsis. If you read the abstract, it is obvious that this second species is brought up with no explanation for why it was examined. Was this species used as a comparison? Was this species based on its abundance, or functional similarity to the target species? None of this is clear, and this part of the abstract appears to be left unchanged from the previous version. In the response to comments document, the authors provide a very vague explanation/response, which does not provide any further information. As a result, my original criticism stands unchanged. There are many other similar examples to this. Furthermore, there are several cases where the authors provide a response in the comments, but no significant changes appear in the main text. This is also a big problem.
To provide another example, I commented previously that the authors should not state that DBH is "the most important parameter used in long-term monitoring." What if someone is monitoring seedling height, or number of leaves, or herbivore damage, etc. In these cases, DBH is not going to be the most ipmortant parameter. There is no need to make this statement, and the authors should remove it. They also do not need to provide such detailed information about what DBH is and why/how it is used. This is a forestry journal, all readers will be familiar with DBH. Unfortunately, the authors simply state that they disagree with me and have decided to keep all of this information in the manuscript.
In addition to the more general organizational problems, the writing itself is still very far from adequate for publication. These problems go beyond simple grammatical errors. A brief skim of the abstract or introduction would provide evidence for these problems.
As mentioned above, this manuscript required very considerable changes to be considered for publication at Forests. I unfortunately do not think these changes have been made by the authors.
Author Response
Response to Reviewer 1 Comments
Point 1: The authors of this manuscript have made many important and significant improvements to their manuscript. At the same time, my previous review of this work indicated that very significant improvements in the writing and focus of the manuscript would be necessary for this to be considered more seriously for publication. While I think the authors have indeed made important steps in the right direction, I still do not find the manuscript to be suitable for consideration at Forests.
Response: Thank you for your important comments. We have revised the text based on the reviewers’ comments, to the extent possible. We would be grateful if you consider our responses and revisions. In addition, we have inserted new text into the abstract of the revised manuscript.
Point 2: One of my most important suggestions was that the authors should add clear research questions/hypotheses, and also provide further details on the motivations behind their work. Unfortunately, the authors seem to have ignored this comment altogether. As a result, the manuscript still severely lacks focus and organization. This is evident throughout the entire text, from introduction through conclusions. If there were two or three clear research questions presented at the end of the introduction, then the entire rest of the manuscript could be organized around these questions. This would admittedly be a lot of work, but I do find this to be necessary. As it stands, the work still seems highly descriptive and extremely focused on the single study site that the authors collected data from. As a result, it is not at all clear what value this work would have to the general readership of Forests.
Response: We have now added hypotheses, and have discussed related theories, based on the advice and opinions presented in the initial reviews. We have addressed a clearer set of study questions, which we phrase as objectives, as follows:
“In the present study, Abies nephrolepis seedlings and other vegetation in the coniferous forests of Seorak Mountain were monitored for 2 years. Accordingly, the first objective of this study was to investigate the spatial and temporal patterns of the dynamics of Abies nephrolepis seedlings in sub-alpine coniferous forests. Based on the studies cited above, which suggest the hypothesis that seedling density is related to environmental factors, our second objective was to assess whether Abies nephrolepis seedlings density is related to biotic and abiotic environmental factors.”
Seorak Mountain has mostly alpine plants, with a wide distribution of dwarf trees. The Abies nephrolepis community occupies a larger area than other species. Therefore, during site selection, we found that this area satisfied our research criteria. As a result, we concentrated on a single site in this region.
Point 3: While the authors did a good job at responding to my comments at length, I often found their responses to either be confusing, or inadequate for the most part. As one example, I asked the authors to explain why they bring up Pinus koraiensis in the abstract and elsewhere, when the title and beginning of abstract make it clear that the study focuses on Pinus nephrolepsis. If you read the abstract, it is obvious that this second species is brought up with no explanation for why it was examined. Was this species used as a comparison? Was this species based on its abundance, or functional similarity to the target species? None of this is clear, and this part of the abstract appears to be left unchanged from the previous version. In the response to comments document, the authors provide a very vague explanation/response, which does not provide any further information. As a result, my original criticism stands unchanged. There are many other similar examples to this. Furthermore, there are several cases where the authors provide a response in the comments, but no significant changes appear in the main text. This is also a big problem.
Response: Thank you for this comment. We have revised the abstract and main text according to these comments. We have added this sentence:
“The majority of Abies nephrolepis had diameter at breast height less than 10 cm, and the number of seedlings increased over the study period. This reflects survival and growth due to successive annual mast seeding events.”
We have added this sentence to the Results:
“At the site, the distributions of DBH for P. koraiensis and Quercus mongolica were bell shaped, whereas those for Abies nephrolepis, Betula ermanii, and 15 other taxa were inverse J shaped (P < 0.001; Figure 3).”
We apologize that we did not adequately explain our inclusion of pine and oak trees in our study. Pine and oak trees coexist at the site. Therefore, we felt that it was important to include them, to reflect the complete population structure of the site. This is because we wanted to assess whether population structure affects seedling density. Our results for the effects of tree basal area support this part of the analysis.
Point 4: To provide another example, I commented previously that the authors should not state that DBH is "the most important parameter used in long-term monitoring." What if someone is monitoring seedling height, or number of leaves, or herbivore damage, etc. In these cases, DBH is not going to be the most ipmortant parameter. There is no need to make this statement, and the authors should remove it. They also do not need to provide such detailed information about what DBH is and why/how it is used. This is a forestry journal, all readers will be familiar with DBH. Unfortunately, the authors simply state that they disagree with me and have decided to keep all of this information in the manuscript.
Response: We have now revised the introduction and the contents of the text accordingly. The modified sentence in introduction is as follows: “Long-term monitoring implies a 5- or 10-year interval between study periods; it is thus difficult to examine continuous forest ecosystem dynamic changes during shorter period [17-21]. Conversely, there are many quantitative studies on the relationships between seed production and climate, topography, and biological factors, but these all have short observation periods, mostly of one year [22]. Many single-site time series studies have proved useful to understand ecosystems. Such single-site time series can be used to observe and understand patterns of natural recruitment.”
Point 5: In addition to the more general organizational problems, the writing itself is still very far from adequate for publication. These problems go beyond simple grammatical errors. A brief skim of the abstract or introduction would provide evidence for these problems.
As mentioned above, this manuscript required very considerable changes to be considered for publication at Forests. I unfortunately do not think these changes have been made by the authors.
Response: We have had the text edited and proof-read by a native English speaker.

Reviewer 2 Report
Thank you very much for the new version of the paper. I see all the suggestions have been adressed thus the paper is ready for publication.
Author Response
Response to Reviewer 2 Comments
Thank you very much for the new version of the paper. I see all the suggestions have been adressed thus the paper is ready for publication.
Response: Thank you for reviewing our paper.
